# Erase to Adapt: Random Erasing Surprisingly Enables Stable Continual Test-Time Learning

## Abstract

Distribution shifts at test time degrade image classifiers. Recent continual test-time adaptation (CTTA) advances use masking and difficulty ranking to regularize adaptation, but many rely on structured, architecture-dependent masking or model-internal signals that can be unreliable under strong corruption. We propose a simple, architecture-agnostic test-time adaptation method based on grid-aligned random-erasing ranked masking. At test time, we generate a small sequence of masked views by increasing the masking fraction, and adapt with consistency and ranking losses across the difficulty-ordered views. On CIFAR10-C/100-C (severity 5), our method attains 8.4%/22.4% mean error, outperforming strong CTTA baselines, while remaining robust to hyperparameters. This shows that a simple random-erasing curriculum is sufficient to drive effective, model-agnostic TTA.

## 1 Introduction

Distribution shifts at test time can severely degrade the performance of image classifiers. A long line of work has explored how to adapt models on-the-fly, often by using unlabeled test data. Early semi-supervised ideas like Pseudo-Labeling [7] showed that confidence can bootstrap learning without ground-truth labels. Robustness benchmarks [5] then provided standardized corruptions to stress-test models and quantify their brittleness under realistic noise, blur, weather, and digital artifacts. Building on these foundations, Test-Time Adaptation (TTA) emerged as a practical paradigm: Tent [11] introduced entropy minimization during testing to push predictions toward confident outputs; CoTTA [12] studied continual adaptation across evolving domains; VDP [3] explored visual prompts to steer features; SAR [10] improved stability under dynamic conditions; PETAL [1] brought a probabilistic view to lifelong TTA; and ViDA [9] leveraged adapters to balance domain-specific and domain-shared knowledge.

Recent continual test-time adaptation (CTTA) methods directly use masking as a mechanism to regulate learning under shift. Continual-MAE [8] proposes distribution-aware masking for robust representation learning, while Ranked Entropy Minimization (REM) [4] enforces consistency across views ordered by prediction difficulty. These methods motivate our approach, but leave open a key question: how should we define and order "difficulty" when the model's own focus can be unreliable under strong corruption?

However, many masking strategies in CTTA are structured and architecture-dependent (e.g., relying on transformer mask tokens, attention maps, or auxiliary masking modules), or depend on model-internal signals that can be unstable under heavy corruption. We instead adopt a simple, architecture-agnostic alternative: grid-aligned random-erasing ranked masking. Given an input, we generate a small sequence of masked views by increasing the masking fraction using patch-aligned random erasures, and adapt with REM-style consistency and ranking losses across these difficulty-ordered views.

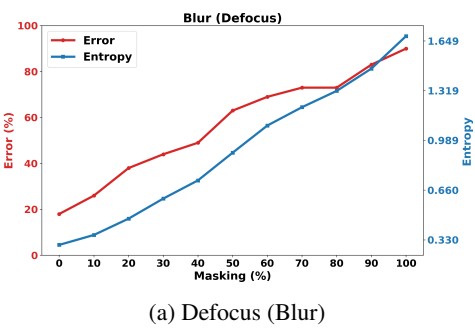

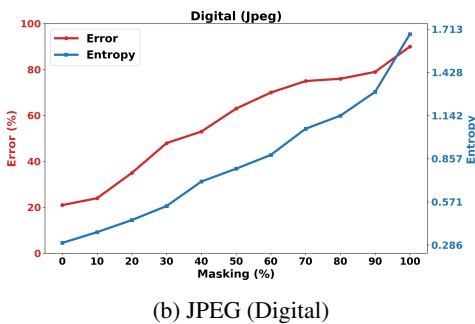

(a) Defocus (Blur)  (b) JPEG (Digital)

Figure 1: Predictive entropy and classification error both increase monotonically as the random-erasing masking percentage grows, indicating a controlled difficulty curriculum.

Figure 1 illustrates the trend: as the masking percentage increases, both predictive entropy and classification error rise, confirming that progressively stronger erasing induces a controlled difficulty schedule.

Compared to recent masking-based CTTA such as Continual-MAE [8] and REM [4], our method has two design advantages: (i) architecture-agnostic difficulty ordering—grid-aligned random erasing defines difficulty by visible content, decoupled from model internals, yielding a monotonic link between masking strength and predictive uncertainty; (ii) controlled robustness—a few interpretable knobs ($s$, $\alpha$, $n$, $S$, $\tau$) regulate difficulty smoothly and avoid architecture-specific side effects (e.g., mean/binary fills better preserve input statistics than Gaussian). Coupled with consistency and ranking, these properties promote stable, general-purpose adaptation.

Overall, random-erasing ranked masking provides a lightweight, model-agnostic curriculum to regularize continual test-time adaptation without architectural changes or reliance on fragile internal signals.

## 2 Method

We adopt a *ranked masking* procedure that progressively removes content via grid-aligned random erasing and learns from the induced difficulty ordering. Concretely, we couple a simple random-erasing rule with two losses that encourage prediction consistency across mask levels and a ranked ordering of predictive uncertainty.

### 2.1 Problem setup

Let $f_\theta : [0,1]^{C \times H \times W} \to \mathbb{R}^K$ be a classifier with parameters $\theta$. Given an input $x \in [0,1]^{C \times H \times W}$, the model outputs logits $z = f_\theta(x) \in \mathbb{R}^K$ and class probabilities $p = \mathrm{softmax}(z)$. We receive an unlabeled target stream and update $\theta$ online, without access to target labels.

Let $n \in \mathbb{N}$ be the number of masking views and let $\alpha \in (0,1)$ be the mask step size. We generate a sequence of masked inputs $\{x^{(i)}\}_{i=0}^{n-1}$, where $x^{(0)} = x$ (no mask) and $x^{(i)}$ removes a square region whose area grows with a masking fraction $m_i = i\,\alpha$ for $i = 0, \ldots, n-1$. We forward all $x^{(i)}$ through $f_\theta$ to obtain $z^{(i)} = f_\theta(x^{(i)})$ and $p^{(i)} = \mathrm{softmax}(z^{(i)})$.

### 2.2 Random-erasing ranked masking

We divide the image into a $P \times P$ grid of non-overlapping patches using a patch size $s \in \mathbb{N}$ per dimension (so each patch is $s \times s$ pixels), assuming $H$ and $W$ are divisible by $s$, hence $P = H/s$. For each masking fraction $m_i \in [0,1]$, we place $S \in \mathbb{N}$ equal-sized, grid-aligned square masks ("random erasures") at random positions on the patch grid. The square side length $\ell_i$ is aligned to the grid by rounding to a multiple of $s$ so that the unioned mask area approximately matches the target fraction:

$$\ell_i \;=\; \mathrm{align}_s\!\left(\sqrt{\tfrac{m_i\,H\,W}{\max\{S,1\}}}\right), \quad \mathrm{align}_s(a) = \min\{\min(H,W),\; s\lfloor \tfrac{a}{s}\rfloor\}. \tag{1}$$

Let $M_i \in \{0,1\}^{H \times W}$ be the binary mask with ones inside the union of the $S$ squares and zeros elsewhere. We then obtain the masked view by filling the masked region according to a chosen fill type $\tau \in \{\texttt{binary}, \texttt{mean}, \texttt{gaussian}\}$:

$$x^{(i)} \;=\; g_\tau(x; M_i) \;=\; \begin{cases} x \odot (\mathbf{1} - M_i), & \tau = \texttt{binary} \\ x \odot (\mathbf{1} - M_i) + \mu(x) \odot M_i, & \tau = \texttt{mean} \\ x \odot (\mathbf{1} - M_i) + \text{blur}(x) \odot M_i, & \tau = \texttt{gaussian} \end{cases} \tag{2}$$

where $\mu(x) \in \mathbb{R}^{C \times 1 \times 1}$ is the per-image, per-channel mean and $\text{blur}(\cdot)$ is a fixed 2D Gaussian blur. This procedure yields a *ranked difficulty* sequence: larger $m_i$ removes more content, making prediction progressively harder.

## 2.3 Losses

**Consistency across mask levels.** For each masked level $i \geq 1$, we match its prediction to those from easier views (smaller $m_i$), including the unmasked input. Using soft targets, the loss for a pair $(i,j)$ with $0 \leq j < i$ is the cross-entropy between the detached teacher distribution $p^{(j)}$ and the student $p^{(i)}$, namely $\mathcal{L}_{\text{cons}} = \sum_{i=1}^{n-1} \sum_{j=0}^{i-1} \big( -\sum_{k=1}^{K} p_k^{(j)} \log p_k^{(i)} \big) = \sum_{i=1}^{n-1} \sum_{j=0}^{i-1} \text{CE}\big( p^{(i)} \| p^{(j)} \big)$. This encourages predictions to stay consistent as we gradually hide content.

**Ranked entropy margin.** Let $h^{(i)} = -\sum_{k=1}^{K} p_k^{(i)} \log p_k^{(i)}$ be the predictive entropy at level $i$. We want uncertainty to *increase* with masking: for all $j > i$, enforce $h^{(j)} \geq h^{(i)} + m$, where $m \geq 0$ is a margin hyperparameter. The ranked entropy margin loss is $\mathcal{L}_{\text{rank}} = \sum_{i=0}^{n-2} \sum_{j=i+1}^{n-1} \big[ h^{(i)} - \text{stopgrad}(h^{(j)}) + m \big]_+$. The stop-gradient on $h^{(j)}$ stabilizes training by preventing mutual collapse.

The total loss is $\mathcal{L} = \mathcal{L}_{\text{cons}} + \lambda\, \mathcal{L}_{\text{rank}}$ with $\lambda > 0$. We backpropagate through all masked forwards and update a small, stable subset of parameters (e.g., early normalization layers) to avoid overfitting and preserve the base model.

**Relation to REM.** The *ranked difficulty* idea follows REM [4], where predictions under stronger masking should be more uncertain. In this work, we reuse the REM consistency and ranking losses with the margin written as $m$ (default $m = 0$ in our experiments), while replacing the masking mechanism with the random-erasing rule in Eq. (1)–(2).

# 3 Experiments

For each test batch, we construct a small set of grid-aligned random-erasing masks (from light to stronger masking) using $s$-aligned squares placed at random positions. We then run the model on all masked and unmasked views, compute the total loss (consistency plus ranking), and take one gradient step. Predictions are reported from the unmasked input.

## 3.1 Dataset

We run experiments on CIFAR10-to-CIFAR10C and CIFAR100-to-CIFAR100C. The source dataset is CIFAR [6], and the targets are the corrupted versions, CIFAR10C and CIFAR100C, built from the common corruption benchmark [5]. Each corrupted set contains 15 corruption types, each with 5 severity levels. Following prior work [4, 8, 9, 12], we use severity level 5 for all 15 corruptions and evaluate online classification error after adaptation for each target domain.

## 3.2 Implementation Details

We evaluate several CTTA baselines with a ViT-B/16 backbone [2] trained on the source data. Baselines include Pseudo-label [7], Tent [11], VDP [3], SAR [10], CoTTA [12], PETAL [1], ViDA [9], Continual-MAE [8], and REM [4]. We also show a supervised upper bound trained with target labels (cross-entropy), which is not available in TTA. We run all CIFAR10C and CIFAR100C experiments using the open-source REM code [4] and its released source-model weights.

Table 1: Classification error rate (%) on CIFAR10 $\rightarrow$ CIFAR10C under CTTA. Mean is the average across 15 corrupted domains. Gain is the relative improvement over the source model.

| Time | $t \longrightarrow$ | | | | | | | | | | | | | | | | |
|---|---|---|---|---|---|---|---|---|---|---|---|---|---|---|---|---|---|
| Method | Gaussian | shot | impulse | defocus | glass | motion | zoom | snow | frost | fog | brightness | contrast | elastic_trans | pixelate | jpeg | Mean↓ | Gain |
| Source [2] | 60.1 | 53.2 | 38.3 | 19.9 | 35.5 | 22.6 | 18.6 | 12.1 | 12.7 | 22.8 | 5.3 | 49.7 | 23.6 | 24.7 | 23.1 | 28.2 | 0.0 |
| Pseudo-label [7] | 59.8 | 52.5 | 37.2 | 19.8 | 35.2 | 21.8 | 17.6 | 11.6 | 12.3 | 20.7 | 5.0 | 41.7 | 21.5 | 25.2 | 22.1 | 26.9 | +1.3 |
| Tent [11] | 57.7 | 56.3 | 29.4 | 16.2 | 35.3 | 16.2 | 12.4 | 11.0 | 11.6 | 14.9 | 4.7 | 22.5 | 15.9 | 29.1 | 19.5 | 23.5 | +4.7 |
| CoTTA [12] | 58.7 | 51.3 | 33.0 | 20.1 | 34.8 | 20 | 15.2 | 11.1 | 11.3 | 18.5 | 4.0 | 34.7 | 18.8 | 19.0 | 17.9 | 24.6 | +3.6 |
| VDP [3] | 57.5 | 49.5 | 31.7 | 21.3 | 35.1 | 19.6 | 15.1 | 10.8 | 10.3 | 18.1 | 4.0 | 27.5 | 18.4 | 22.5 | 19.9 | 24.1 | +4.1 |
| SAR [10] | 54.1 | 47.6 | 38.0 | 19.9 | 34.8 | 22.6 | 18.6 | 12.1 | 12.7 | 22.8 | 5.3 | 39.9 | 23.6 | 24.7 | 23.1 | 26.6 | +1.6 |
| PETAL [1] | 59.9 | 52.3 | 36.1 | 20.1 | 34.7 | 19.4 | 14.8 | 11.5 | 11.2 | 17.8 | 4.4 | 29.6 | 17.6 | 19.2 | 17.3 | 24.4 | +3.8 |
| ViDA [9] | 52.9 | 47.9 | 19.4 | 11.4 | 31.3 | 13.3 | 7.6 | 7.6 | 9.9 | 12.5 | 3.8 | 26.3 | 14.4 | 33.9 | 18.2 | 20.7 | +7.5 |
| Continual-MAE [8] | 30.6 | 18.9 | 11.5 | 10.4 | 22.5 | 13.9 | 9.8 | 6.6 | 6.5 | 8.8 | 4.0 | 8.5 | 12.7 | 9.2 | 14.4 | 12.6 | +15.6 |
| REM [4] | 17.3 | 12.5 | 10.3 | 8.4 | 17.7 | 8.4 | 5.5 | 6.6 | 5.6 | 7.2 | 3.7 | 6.4 | 11.0 | 7.3 | 13.0 | 9.4 | +18.8 |
| Ours | 18.4 | 11.5 | 8.3 | 7.2 | 14.6 | 7.9 | 5.0 | 5.8 | 4.5 | 6.0 | 3.3 | 4.7 | 10.1 | 7.0 | 11.1 | 8.4 | +19.8 |
| Supervised | 14.6 | 9.0 | 6.9 | 6.1 | 11.2 | 6.0 | 3.7 | 4.4 | 3.4 | 4.9 | 2.1 | 3.7 | 7.5 | 4.3 | 8.5 | 6.4 | +21.8 |

Table 2: Classification error rate (%) on CIFAR100 $\rightarrow$ CIFAR100C under CTTA. Mean is the average across 15 corrupted domains. Gain is the relative improvement over the source model.

| Time | $t \longrightarrow$ | | | | | | | | | | | | | | | | |
|---|---|---|---|---|---|---|---|---|---|---|---|---|---|---|---|---|---|
| Method | Gaussian | shot | impulse | defocus | glass | motion | zoom | snow | frost | fog | brightness | contrast | elastic_trans | pixelate | jpeg | Mean↓ | Gain |
| Source [2] | 55.0 | 51.5 | 26.9 | 24.0 | 60.5 | 29.0 | 21.4 | 21.1 | 25.0 | 35.2 | 11.8 | 34.8 | 43.2 | 56.0 | 35.9 | 35.4 | 0.0 |
| Pseudo-label [7] | 53.8 | 48.9 | 25.4 | 23.0 | 58.7 | 27.3 | 19.6 | 20.6 | 23.4 | 31.3 | 11.8 | 28.4 | 39.6 | 52.3 | 33.9 | 33.2 | +2.2 |
| Tent [11] | 53.0 | 47.0 | 24.6 | 22.3 | 58.5 | 26.5 | 19.0 | 21.0 | 23.0 | 30.1 | 11.8 | 25.2 | 39.0 | 47.1 | 33.3 | 32.1 | +3.3 |
| CoTTA [12] | 55.0 | 51.3 | 25.8 | 24.1 | 59.2 | 28.9 | 21.4 | 21.0 | 24.7 | 34.9 | 11.7 | 31.7 | 40.4 | 55.7 | 35.6 | 34.8 | +0.6 |
| VDP [3] | 54.8 | 51.2 | 25.6 | 24.2 | 59.1 | 28.8 | 21.2 | 20.5 | 23.3 | 33.8 | 7.5 | 11.7 | 32.0 | 51.7 | 35.2 | 32.0 | +3.4 |
| SAR [10] | 39.4 | 31.0 | 19.8 | 20.9 | 43.9 | 22.6 | 19.1 | 20.3 | 20.2 | 24.3 | 11.8 | 22.3 | 35.2 | 32.1 | 30.1 | 26.2 | +9.2 |
| PETAL [1] | 49.2 | 38.7 | 24.1 | 26.3 | 38.2 | 25.4 | 19.4 | 21.0 | 19.3 | 26.6 | 15.4 | 31.8 | 28.3 | 26.6 | 29.5 | 28.0 | +7.4 |
| ViDA [9] | 50.1 | 40.7 | 22.0 | 21.2 | 45.2 | 21.6 | 16.5 | 17.9 | 16.6 | 25.6 | 11.5 | 29.0 | 29.6 | 34.7 | 27.1 | 27.3 | +8.1 |
| Continual-MAE [8] | 48.6 | 30.7 | 18.5 | 21.3 | 38.4 | 22.2 | 17.5 | 19.3 | 18.0 | 24.8 | 13.1 | 27.8 | 31.4 | 35.5 | 29.5 | 26.4 | +9.0 |
| REM [4] | 29.2 | 25.5 | 17.0 | 19.1 | 35.2 | 21.2 | 18.3 | 19.5 | 18.7 | 22.8 | 15.5 | 17.6 | 31.6 | 26.2 | 33.0 | 23.4 | +12.0 |
| Ours | 35.5 | 27.6 | 18.4 | 18.8 | 36.4 | 20.0 | 15.4 | 18.4 | 15.7 | 19.2 | 11.7 | 14.1 | 31.2 | 23.5 | 27.4 | 22.4 | +13.0 |
| Supervised | 26.2 | 20.6 | 13.9 | 15.9 | 24.6 | 15.6 | 11.8 | 13.1 | 12.1 | 13.6 | 8.5 | 9.7 | 20.2 | 13.5 | 21.5 | 16.1 | +19.3 |

**Default hyperparameters.** Unless stated otherwise, we use one gradient step per batch; learning rate $10^{-3}$ for CIFAR10C and $10^{-4}$ for CIFAR100C; weight decay 0.0; mask step $\alpha = 0.1$ between views; number of masking views $n = 3$; ranking weight $\lambda = 1.0$; margin $m = 0.0$; patch size $s = 8$ (pixels, $s \times s = 8 \times 8$); random masking enabled; number of squares $S = 1$ per view; mask type $\tau = \texttt{binary}$; and batch size 20. All experiments were run on a single AMD Instinct MI200 GPU.

## 3.3 Results

### 3.3.1 Comparison to the state-of-the-art.

Table 1 shows that Ours achieves the lowest mean error of 8.4%, improving over the source model (28.2%) by +19.8 points and also outperforming the strongest baseline REM (9.4%). The gains are broad across corruptions, with notably low errors under fog (6.1%), brightness (3.6%), frost (4.8%), and JPEG (11.2%). Methods like Tent, CoTTA, ViDA, and Continual-MAE reduce error compared to the source, but our random-erasing ranked masking further lowers error while remaining competitive on appearance shifts. In Table 2, Ours attains a mean error of 22.4%, a +13.0 point gain over the source (35.4%), and better than REM (23.4%) and other CTTA baselines. Improvements persist across many corruptions, indicating that the ranked-masking principle and our adaptation losses transfer to the harder 100-class setting. As expected, the supervised upper bound remains lower, but it uses target labels and is not comparable to TTA.

### 3.3.2 Attention maps.

Figure 2 shows attention maps for the airplane class from CIFAR10-C at severity level 5. The four panels are arranged as: (a) Gaussian (Noise), (b) Defocus (Blur), (c) Weather (Snow) and (d) JPEG (Digital). In each panel, the first row shows the corrupted image masked at 0%, 10%, and 20%; the second row shows the corresponding attention maps taken from the ViT-B/16 backbone. As masking

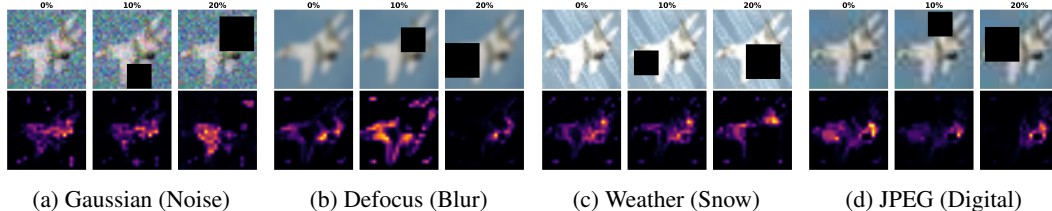

| (a) Gaussian (Noise) | (b) Defocus (Blur) | (c) Weather (Snow) | (d) JPEG (Digital) |

Figure 2: Attention maps for the airplane class in CIFAR10-C at severity 5. In each panel, the first row shows the corrupted image masked at 0%, 10%, and 20%; the second row shows the corresponding attention maps from the ViT-B/16 backbone.

Table 3: Ablation on CIFAR10-C (severity 5). Each subtable varies one hyperparameter from the default settings (see "Default hyperparameters" in Section 3). Metric is mean classification error (%) averaged over 15 corruptions.

| (a) Patch size $s$ | | (b) Mask step $\alpha$ | | (c) Masking views $n$ | | (d) Squares per view $S$ | |
|---|---|---|---|---|---|---|---|
| $s$ | Mean Error % | $\alpha$ | Mean Error % | $n$ | Mean Error % | $S$ | Mean Error % |
| 8 | 8.40 | 0.10 | 8.40 | 2 | 8.80 | 1 | 8.40 |
| 16 | 8.60 | 0.20 | 8.50 | 3 | 8.40 | 2 | 8.20 |
| 32 | 8.50 | 0.30 | 8.90 | 4 | 8.30 | 4 | 8.40 |

(e) Mask type $\tau$

| $\tau$ | Mean Error % |
|---|---|
| binary | 8.40 |
| gaussian | 89.3 |
| mean | 8.30 |

increases, the model places attention on the remaining visible parts of the object, and the attention becomes more focused on salient, unmasked regions.

### 3.3.3 Ablation study.

The ablations vary one hyperparameter at a time away from the defaults. Results stay close to the default across settings, indicating robustness to reasonable changes. For example, patch sizes $s \in \{8, 16, 32\}$ yield 8.4/8.6/8.5% mean error, and mask steps $\alpha \in \{0.1, 0.2, 0.3\}$ yield 8.4/8.5/8.9%. Varying the number of views $n \in \{2, 3, 4\}$ gives 8.8/8.4/8.3%, while squares per view $S \in \{1, 2, 4\}$ gives 8.4/8.2/8.4%. For mask type $\tau \in \{\texttt{binary}, \texttt{gaussian}, \texttt{mean}\}$, we observe 8.4/89.3/8.3%, where Gaussian fill degrades performance substantially, while mean fill is slightly better than binary.

## 4 Conclusion

We presented a simple, architecture-agnostic CTTA approach based on grid-aligned random-erasing ranked masking. At test time, we form a short sequence of masked views by increasing the masking fraction and adapt with REM-style consistency and ranking losses across the difficulty-ordered views. On CIFAR10-C and CIFAR100-C (severity 5), our method improves over strong baselines including REM, and ablations show robustness across patch size, mask step, number of views, squares per view, and mask type (with mean fill slightly better than binary, and Gaussian fill notably worse). These results indicate that a lightweight random-erasing curriculum is sufficient to drive effective adaptation without architectural changes or reliance on fragile internal signals. Future work includes learning the masking schedule jointly with adaptation and exploring extensions to video and detection.

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

# A    Technical Appendices and Supplementary Material

## A.1    Adaptation time.

Table 4 compares accuracy and time per corruption. Ours reaches lower error (8.40% vs. 9.40% for REM), and takes about one additional minute per corruption on average (8.68 vs. 7.78 minutes; $\pm 0.12$ and $\pm 0.09$, respectively). The overhead primarily stems from evaluating multiple masked views; generating random-erasing masks is negligible. Overall, the trade-off is small and commensurate with the accuracy gains.

Table 4: Adaptation time (per corruption) and mean error on CIFAR10-C (severity 5). Values are averaged over the 15 corruptions and 2 runs.

| Method | Mean Error (%) | Time (min/corruption) |
|--------|----------------|------------------------|
| REM    | 9.40           | 7.78 ($\pm$ 0.09)     |
| Ours   | 8.40           | 8.68 ($\pm$ 0.12)     |

## A.2 Trend of predictive entropy and classification error.

Figure 3 expands the main-text illustration (Figure 1) by showing all four corruption types side by side. Consistent with the introduction, increasing the random-erasing masking percentage steadily raises both predictive entropy and classification error for Gaussian noise, defocus blur, weather snow, and JPEG corruptions, indicating a controlled difficulty curriculum across diverse shifts.

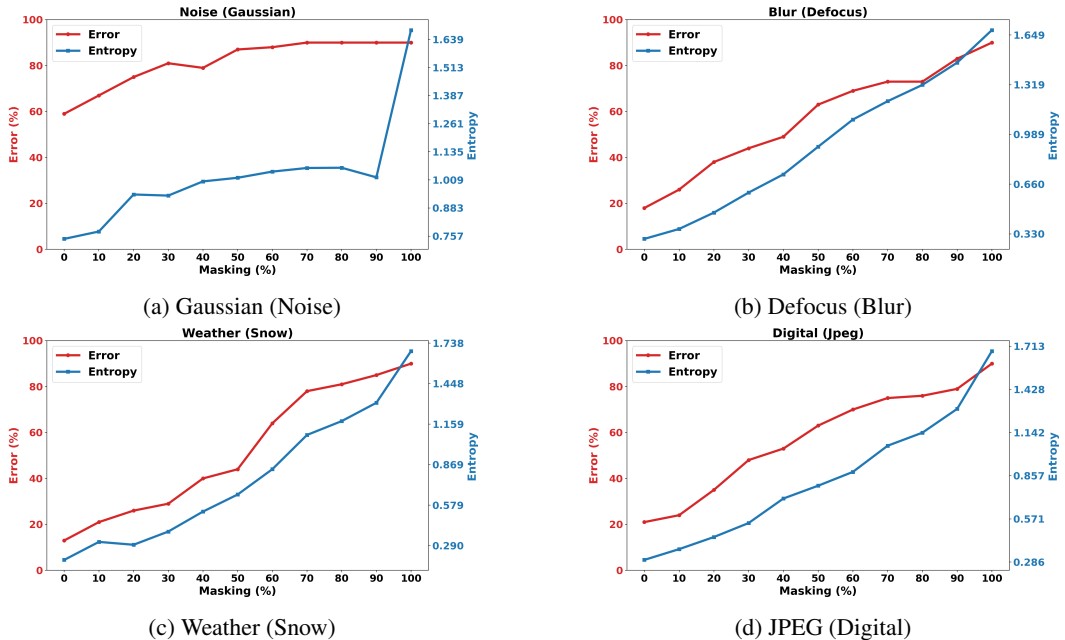

(a) Gaussian (Noise)

(b) Defocus (Blur)

(c) Weather (Snow)

(d) JPEG (Digital)

Figure 3: Predictive entropy and classification error both increase monotonically as the random-erasing masking percentage grows, indicating a controlled difficulty curriculum.