# OpenReview forum: "Erase to Adapt: Random Erasing Surprisingly Enables Stable Continual Test-Time Learning"
_EurIPS.cc/2025/Workshop/UPLB — UPLB2025_

### Official Review · Reviewer_Hi5B · 2025-10-27
**This paper presents a new architecture-agnostic method for continual test-time adaptation (CTTA), based on random erasing masks. Through a systematic comparison comparison with previously existing CTTA frameworks on different tasks, the authors empirically show a consistent reduction of the classification error rate.**

**Rating:** 7
**Confidence:** 3

**Review:**

The paper is well structured and clear, making it easy to follow. In the introduction, the problem of CTTA is presented in a concise, yet exhaustive way by making reference to the relevant literature. The theory behind the new methodology is explained in Sect. 2 (Methods) and it is mathematically sound. The new approach seem to me a slight modification of the Ranked Entropy Minimization (REM), but its effectiveness is clearly shown in the results section, where the authors show a clear gain with respect to previous architecture-dependent methods. However, the difference between this and REM does not seem huge.

Pros:
- Well structured, clear, original idea
- All information for reproducibility are included in the paper
- It fits the topic of the workshop as a method to tackle distribution shifts.
- Exhaustive comparisons with different methods and and tasks from the literature

Cons
- It is a slight modification of a recently published method. It fits as a workshop paper but it may need more work to be presented as a full, new original publication
- This work is only experimental and lacks of a theoretical understanding of why it works.
- The gain with respect REM is not big and not consistently better across tasks.
- Experiments are limited to a VIT-B/16 architecture and only on CIFAR10/100 datasets.

Overall, this work presents an original idea that fits the purpose of this workshop. Though it may need more refinement as a future publication, it may stimulate interesting discussions in the venue.

---

### Decision · Program_Chairs · 2025-11-03

Accept (Poster)